## Research Article

community-initiated care; community-based care; task-sharing; psychosocial interventions; mental health

**Corresponding author:**
Vikram Patel;
Email: vikram_patel@hms.harvard.edu

# A theory of change for community-initiated mental health care in the United States

Erica Breuer[1], Angelika Morris[2], Laura Blanke[3], Miriam Pearsall[4], Roxana Rodriguez[3], Benjamin F. Miller[5], John A. Naslund[2], Shekhar Saxena[2], Satchit Balsari[2,6] and Vikram Patel[2]

[1]College of Health, Medicine and Wellbeing, University of Newcastle, Newcastle, NSW, Australia; [2]Department of Global Health and Social Medicine, Harvard Medical School, Boston, MA, USA; [3]Well Being Trust, Oakland, CA, USA; [4]National Academy for State Health Policy (NASHP), Portland, ME, USA; [5]Department of Psychiatry and Behavioural Sciences, Stanford School of Medicine, Palo Alto, CA, USA and [6]Emergency Medicine, Beth Israel Deaconess Medical Center, Boston, MA, USA

## Abstract

Mental health service delivery needs radical reimagination in the United States where unmet needs for care remain large and most metrics on the burden of mental health problems have worsened, despite significant numbers of mental health professionals, spending on service provision and research. The COVID-19 pandemic has exacerbated the need for mental health care. One path to a radical reimagination is "Community Initiated Care (CIC)" which equips and empowers communities to address by providing brief psychosocial interventions by people in community settings. We co-developed a theory of change (ToC) for CIC with 24 stakeholders including representatives from community-based, advocacy, philanthropic and faith-based organizations to understand how CIC could be developed and adapted for specific contexts. We present a ToC which describes ways in which the CIC initiative can promote and strengthen mental health in communities in the United States with respect to community organization and leadership; community care and inclusion and normalizing mental health. We propose 10 strategies as part of CIC and propose a way forward for implementation and evaluation. This CIC model is a local, tailored approach which can expand the role of community members to strengthen our response to mental health needs in the United States.

## Impact statement

The American mental health system is in crisis despite high levels of spending and highly trained health professionals. This has been exacerbated by the COVID-19 pandemic. With an estimated 21% of Americans experiencing mental illness in 2020 and only about a one-fifth are likely to be receiving adequate care, we need to think creatively about how to solve this problem (2, 5, 6). In this article, we build on the strong and growing evidence for the effective intervention delivery by lay health workers and community members and propose an initiative called Community Initiated Care (CIC) which will empower communities to be able to address mental health problems (24). We have operationalized CIC with stakeholders and present a detailed theory of change for how we propose CIC will work. This first step lays the groundwork for the further operationalization and implementation of this initiative which is likely to be both cost-effective and scalable and can be adapted, implemented and evaluated in a variety of settings in the United States.

## Background

Mental health, comprising emotional, psychological and social well-being, affects how humans think, feel and act. Mental health problems affect people's emotions, cognition and behavior and can range from nonspecific mental distress to formally diagnosed mental illnesses (Patel et al., 2018). Mental illness is a large contributor to the global burden of disease (Whiteford et al., 2016). An estimated 21% of adults (52.9 million people) in the United States experienced mental illness in 2020 (Substance Abuse and Mental Health Services Administration, 2021). Mental health care, as understood and practiced in contemporary organized health systems, including in the United States, has largely followed a biomedical approach where care is typically contingent on a clinical diagnosis and is the domain of specialist mental health providers. These include psychiatrists and psychologists who provide a mix of expensive outpatient and inpatient care with substantial out of pocket costs (Sundararaman, 2009). The United States has about 145 mental health workers per 100,000 population (compared to a global average of 13) (World Health Organization, 2021a, 2021c) which is unevenly geographically spread and poorly organized (Tikkanen et al., 2020).

Annual health spending on mental health problems in the United States was estimated to increase from $171.7 billion in 2009 to more than $250 billion in 2019 (Substance Abuse and Mental Health Services Administration, 2014). Despite significant spending on mental health care which is estimated to exceed that for any other health condition (Roehrig, 2016), only about 42% of people with mental health problems had access to any treatment prior to the COVID-19 pandemic (Evans-Lacko et al., 2018), with only about half of these likely to be accessing adequate care (Vigo et al., 2020). Access to care is worse for many historically disadvantaged populations and groups and in rural areas (Wielen et al., 2015). For example, Black, Indigenous and people of color experience substantive additional barriers to both identification (Kato et al., 2018) and treatment of mental health problems (Derr, 2016) and health outcomes (Friedman and Hansen, 2022). Lack of insurance coverage, limited access to specialist providers, overburdened hospitals, fragmented service delivery models, and the high cost of care had already made mental health care inaccessible for many Americans (Saechao et al., 2012). The pandemic has deepened this crisis, recently described as a "crisis of care" (Insel, 2022), with inequitable impact on racial minorities, the economically disadvantaged and women (Collaborators, 2021; Islam et al., 2021; Friedman and Hansen, 2022; Thomeer et al., 2023).

The current biomedical approach to care views mental health problems through a diagnostic prism focused on categorizing symptoms into one or more discrete disorders based on specific symptom thresholds (Regier, 2007). It often ignores the interplay between biological and behavioral determinants, and the complex socioeconomic determinants of mental health (Lund et al., 2018) such as poverty, unemployment, food insecurity, lack of safety, poor housing, social exclusion and structural racism. However, in recent years there has been a move toward dimensional and transdiagnostic approaches to diagnosis and treatment (Dalgleish et al., 2020) with an increased recognition that mental health problems exist on a continuum (Patel et al., 2018). This dimensional approach views many mental illnesses as extreme and stressful versions of common, normative, human experiences (e.g., sadness, loneliness, grief, despair) (Whitley and Drake, 2010) which create sustained dysfunction over time if not addressed. By understanding mental health problems as a series of experiences of varying severity rather than a binary biomedical diagnosis, prevention and care for mental health problems aims to provide better support for people through these experiences, consistent with the goals of prevention and early intervention (Patel et al., 2018).

Specialist mental health providers alone cannot realistically, or cost-efficiently, meet the population need for prevention, promotion and early mental health interventions throughout the life course. Indeed, the World Health Organization (WHO), in their Service Organization Pyramid for an Optimal Mix of Services for Mental Health (Funk et al., 2004), describes three interlinked tiers of services: *self-care and informal community healthcare* for all; widely available *primary health care* and *specialist mental health care* when warranted at the top of the pyramid. It calls for the expansion of the role that individuals themselves, families, friends and communities play in providing *self-care and informal community healthcare.*

There is increasing recognition that it is important and effective to equip individuals and communities with strategies to provide timely support to complement the role of (and, where possible, reduce the need for) specialist care (McBain et al., 2021; World Health Organization, 2021b). This support can be provided

through brief, structured psychosocial interventions which are delivered in community settings through "task-sharing" with community-based workers and other community members. Task-sharing involves the collaborative redistribution of health tasks within diverse workforces and with members of the community (Orkin et al., 2021). For example, instead of a psychologist delivering cognitive behavioral therapy, a community health worker is trained to deliver a brief structured intervention comprising the "active ingredients" of cognitive behavioral therapy with appropriate training, supervision and quality management and delivering care within a coordinated, collaborative care model. Task-sharing aligns with the growing policy impetus in the United States to expand the community-based workforce for mental health (The White House, 2022). Recent systematic reviews of task-sharing interventions for mental health problems in low and middle-income countries, offer strong evidence of its acceptability and effectiveness (Padmanathan and De Silva, 2013; van Ginneken et al., 2013; Barbui et al., 2020). Other reviews document similar evidence from the United States Barnett et al. (2018), including 46 trials of task-sharing specifically for perinatal depression in high-income countries (Singla et al., 2021). In addition to the strong evidence in support of the effectiveness of task-sharing of psychosocial interventions for mental health problems, there is modest evidence of the effectiveness of task-sharing for mental health promotion (Galante et al., 2021), and stronger evidence for prevention (in particular indicated prevention) (Stockings et al., 2016; Arango et al., 2018; Castillo et al., 2019; Le et al., 2021).

Strategies for supporting and caring for others have also been argued to have a strong evolutionary imperative (Kohrt et al., 2020). Communities have incentives to help support social repair and community cohesion (Kohrt et al., 2020). Caring for others provides benefits to both the individuals who provide and receive support (Cole et al., 2015) as well as their community (Kohrt et al., 2020). Members of a community have existing ties to each other which may allow them to proactively prevent and promote mental health as well as recognize people in distress (Mendenhall et al., 2014). They often share identity, common social factors and lived experiences, which provide unique perspective and understanding of the specific needs within their community (Mendenhall et al., 2014; Kohrt et al., 2018, 2020). Care in the community is generally less stigmatized than in the formal mental healthcare system (although, admittedly, this is not always the case where emotions may be the result of not aligning with the social norms of the community) (Kohrt et al., 2018, 2020).

Despite the growing evidence on the acceptability and effectiveness of community involvement in mental healthcare, there are few examples of how to implement, scale and sustain these interventions (Siddiqui et al., 2022). We propose Community Initiated Care (CIC) as a model which aims to expand on existing task-sharing strategies by focusing on involving community members rather than frontline health workers (such as peer support workers or community health workers) who have a specific role within existing health system structures (Kohrt et al., 2023). This is informed by existing community-initiated care models implemented both in the United States and elsewhere (Siddiqui et al., 2022). Within this model, we propose that community members or CIC helpers are trained and supported to work within their own communities to identify and reach out to people with mental health problems, combining knowledge of evidence-based psychological treatments together and leveraging social connectedness, with a focus on indicated prevention, early intervention and promoting recovery (Kohrt et al., 2023). The CIC model would

supplement and strengthen rather than replace existing formal mental healthcare. This is especially important for people with severe mental health problems, for example with psychotic symptoms, who may need continuing care from the specialist mental healthcare system; in such instances, CIC would support the person toward long-term recovery goals, for example through facilitating engagement with formal health services and staying connected with their community.

To design, implement and evaluate CIC, it is critical that we have a generalizable framework for this initiative which can be customized across diverse contexts. With this objective, we codeveloped a theory of change (ToC) of how and why the initiative is likely to realize its intended impact is an important first step. In this article, we describe the methods and results of a ToC workshop with stakeholders which illustrates the hypothesized pathway for how we might advance CIC in the United States to address the large, unmet and urgent need for mental health care.

## Methods

### The ToC approach

ToC is an outcomes-based approach which makes explicit the program theory of an program, initiative or policy and how this is hypothesized to reach its impact (De Silva et al., 2014). ToC has been used increasingly in public health interventions (Breuer et al., 2016), and mental health interventions in particular (Fuhr et al., 2020). We used the ToC approach because it is a flexible (Prinsen and Nijhof, 2015) but structured participatory method which involves diverse stakeholders who provide a rich understanding of the context, potential barriers and facilitators, feasibility of the proposed program and promotes buy-in from stakeholders (Breuer et al., 2014). The result of the ToC process is a theory for how the program is hypothesized to work (De Silva et al., 2014) and provides a clear framework which can be used to develop indicators to monitor and evaluate the program (Breuer et al., 2016). Facilitators work with the stakeholders to identify the intended *impact* or long-term vision of a program or policy, the *short-, medium- and long-term outcomes* on the path to impact and *implementation strategies and interventions* (CIC strategies) needed to achieve these outcomes (De Silva et al., 2014). The process also highlights the *assumptions* or conditions which need to be in place for the outcomes to lead to the impact and the *rationale* or evidence underlying how the program will lead to impact and *indicators* to measure whether the outcomes or impact have been achieved. ToC can be informed by existing guidelines, evidence and theories and frameworks (De Silva et al., 2014). Although no formal guidance exists for reporting on the use of ToC, we use items relevant to ToC development from checklist proposed by Breuer et al. (2016).

Below, we describe how we used a five-step ToC development process to develop a ToC for CIC:

1. *ToC workshops*: We conducted two sequential online workshops (due to pandemic travel and meeting restrictions) with 24 stakeholders to develop a draft ToC. We used purposive sampling (Palinkas et al., 2015) to select participants with expertise and engagement with community-based mental health from diverse contexts and sectors settings in the United States. Stakeholders included mental health service providers, researchers, local government officials and representatives from community-based, advocacy, philanthropic and faith-based organizations. During the first workshop stakeholders (1) identified key challenges within the context; (2) refined the impact and (3) mapped outcomes on the pathway to this impact. In the second workshop, stakeholders (4) reviewed a draft ToC map, (5) made explicit the underlying assumptions in the ToC map; (6) identified the strategies necessary to reach the outcomes and impact and (7) made explicit the underlying principles which should guide the CIC. The ToC workshops were structured according to guidelines by Breuer et al. (2019) using Zoom (Zoom Video Communications, 2022) and Mural (Tactivos, 2022), an online collaborative workspace. We used a mix of plenary and facilitated small group sessions enhanced by being able to give written input on the Mural board and on the chat to ensure all participants were able to participate and give feedback. Minor disagreements during the workshops were resolved by discussion. There were no major disagreements. The meetings were recorded, and one team member (A.M.) ensured all the key points were captured on the Mural board which was used for the analysis.

2. *Between- and post-workshop ToC development*: A core ToC team (E.B., A.M., L.B., R.R., M.P., J.A.N.) led the iterative development of the ToC during weekly met in weekly online meetings over 4 months to refine and consolidate the inputs from the ToC workshops. We focused on workshop inputs from each of the sessions, grouped them thematically and refined and/or combined the inputs in Mural to develop the ToC. Where the inputs were out of the scope of the project, we discussed these and reached a group consensus on whether and how to include them. We structured the ToC using the inputs from the workshop and then were also informed by the World Health Organization Service Organization Pyramid for an Optimal Mix of Services for Mental Health (Funk et al., 2004) to initially frame the levels of implementation but adapted these to suit the CIC setting which is largely community-based and outside the formal healthcare system. We adapted the first three temporal phases of implementation from the Quality Implementation Framework (Meyers et al., 2012) to inform the phases of the ToC. We reviewed the ToC against Proctor's implementation, client and service level outcomes to ensure the outcomes relevant to the project were included (Proctor et al., 2009).

3. *Refinement of strategies*: We considered the list of activities suggested by stakeholders by level and phase of implementation as well as how they aligned with the ERIC taxonomy of implementation strategies (Powell et al., 2015) and Leeman and colleagues' classification of implementation strategies (Leeman et al., 2017). Using evidence from a landscape analysis conducted by the EMPOWER group (Siddiqui et al., 2022) and group discussions we refined the identified the phase of the ToC where the activity would take place, the actor and target of the actions (Proctor et al., 2013). We sorted the list of activities suggested by stakeholders by theme, pathway and phase of implementation (Table 1). We then used evidence from the landscape analysis and group discussions to refine these and group them into 10 strategies. For example, under the first strategy, *using evidence, monitoring, evaluation and learning* we included three activities: conduct landscape and legal analysis; partner with scholars to develop foundational key aspects of CIC intervention for pilot sites and identify capacity and develop strategy for collecting information for monitoring, evaluation and continuous quality improvement. Our 10 strategies will be further operationalized as part of CIC.

**Table 1.** CIC strategies

| CIC strategies | Phases | | Pathway | Actor(s) | Target(s) of actions |
|---|---|---|---|---|---|
| 1. Use evidence, monitoring, evaluation and learning | | | | | |
| | a. Conduct landscape and legal analysis | Planning and engaging | 1, 2 | Research and program teams | Funders, decision-makers, implementing organizations, general public |
| | b. Partner with scholars to develop foundational key aspects of CIC intervention for pilot sites | Building capacity and systems | 2 | Program team | Implementing organizations |
| | c. Identify capacity and develop strategy for collecting information for monitoring, evaluation and continuous quality improvement | Implementing the initiative; long-term outcome | 2 | Research and program teams | Implementing organizations |
| 2. Finance and fund CIC | | | | | |
| | a. Collate the evidence to support and articulate the need and potential return on investment | Planning and engaging | 1 | Research and program teams | Funders, decision-makers, implementing organizations |
| | b. Work with funders and decision-makers to seek methods for financing to support the CIC initiative and community-identified needs | | 1 | Program team | Decision-makers, communities |
| | c. Identify needed resources and motivating factors for adopting CIC at the organizational pathway and engaging in training at the individual pathway | Building capacity and systems | 1, 2 | Implementing organizations | CIC helpers, community members |
| 3. Engage stakeholders | | | | | |
| | a. Convene stakeholders that can inform language used for CIC | Planning and engaging | 3 | Program team and implementing organizations | Community members |
| | b. Host conversations/dialog in different community settings | Building capacity and systems | 3 | Program team implementing organizations | Community leaders, implementing organizations, community members |
| | c. Clear communication among stakeholders about CIC helper role | | 1, 3 | Implementing organizations | Community members |
| | d. Dialog with clinicians about the role of CIC helpers | | 3 | Research and program teams | Clinicians |
| 4. Adapt and tailor CIC | | | | | |
| | a. Apply process developed to engage communities in the codesign of localized initiative to meet their needs through respectful dialog and formal partnership with community-based organizations and other relevant stakeholders (including healthcare, faith-based organizations, police, justice and prison system, disadvantaged or marginalized groups as relevant) | Planning and engaging; Building capacity and systems | 1,2 | Program team | Implementing organizations |
| | b. Create setting-specific CIC models, for example for universal social-emotional learning in schools or normalizing mental health in the workplace | | 2 | Research and program teams, implementing organizations | CIC helpers, communities, community members |
| 5. Integrate CIC with existing care provision | | | | | |
| | a. Internalize a process for identifying and developing connections, partnerships and referral pathways with other existing community-based organizations and/or social services | Planning and engaging | 2 | Implementing organizations / CIC helpers | Community members |
| | b. Map out existing community services which can complement CIC | Building capacity and systems | 2 | Implementing organizations | CIC helpers |
| | c. CIC helpers refer those who needs they cannot meet to community services or formal clinical care | Implementing the initiative | 2 | CIC helpers | Community members |

*(Continued)*

**Table 1.** (*Continued*)

| CIC strategies | Phases | Pathway | Actor(s) | Target(s) of actions |
|---|---|---|---|---|
| **6. Build capacity** | | | | |
| | a. Develop, adapt and provide a set of tiered evidence-based skills, competencies and for CIC helpers | Planning and engaging | 2 | Research and program teams | CIC helpers, community members |
| | b. Identify potential CIC helpers within the community | Building capacity and systems | 2 | Implementing organizations | Community members |
| | c. Equip CIC helpers with knowledge and skills through training which is appropriate for their role within the community-based setting | | 2 | Implementing organizations | CIC helpers |
| **7. Provide technical assistance and mentorship** | | | | |
| | a. Develop a central hub for support, supervision, technical assistance (including evaluation) and mentorship | Building capacity and systems | 1 | Program team | Implementing organizations |
| | b. Develop a learning pathway for CIC helpers | | 2 | Research and program teams | CIC helpers |
| | c. Enable communication and conversation about mental health, including between CIC helpers | Implementing the initiative | 1,3 | | CIC helpers |
| **8. Reach out to communities** | | | | |
| | a. Develop a communication toolkit for CIC in consultation with community-based organizations | Planning and engaging | 3 | Program and communications team | Community members |
| | b. Provide actionable steps as part of awareness campaigns | | 3 | | Implementing organizations, community members |
| | c. Develop specific strategies and materials to reach specific populations of the community | | 3 | | Community members |
| | d. Implement the communications toolkit for CIC, inclusive of:<br>– Engaging influencers in the community (including business leaders, spiritual leaders, arts/performers) to increase awareness<br>– Leveraging social media to increase awareness<br>– Use technology to support self-help and access to CIC | Implementing the initiative | 3 | Programs team, CBO | Implementing organizations, community members |
| **9. Support leadership** | | | | |
| | a. Leadership training to underscore need for this approach, training and tools (TA) to adopt, infusing training into organizational practices | Building capacity and systems | 1 | Research and program teams | National, state and local leaders, implementing organizations |
| **10. Provide evidence-based care and support** | | | | |
| | a. Develop an evidence-based psychosocial interventions for CIC by applying learning from existing evidence-based community interventions and care provisions | Planning and engaging<br>Planning and engaging | 2 | Research and program teams | CIC helpers |
| | b. Define the role of CIC helpers and boundaries of care | Planning and engaging | 1 | Research and program teams; Implementing organizations | CIC helpers |
| | c. Provide a stepped model of care CIC flexibly with respect to time and place in line with the needs of the community, which is equitable, safe, individualized, contextually appropriate for people in need of support | Implementing the initiative | 2 | Implementing organizations, CIC helpers | Community members |

CIC, Community Initiated Care; Pathway 1, community organization and leadership; Pathway 2, community-initiated care; Pathway 3, inclusion and normalizing mental health.

4. *Refinement of indicators*: To develop indicators for each of the ToC outcomes, we identified one or more constructs we needed to measure, the level of analysis (organizational provider, individual provider, individual consumer, community member), phase of data collection, the type of data collection and measurement instrument. We also included key process indicators for each phase of implementation. These will be refined further for each context.

5. *Final presentation of ToC to stakeholders*: Finally, we abbreviated the ToC and presented it to the 24 stakeholders who were part of the initial ToC workshops, as well as an additional 10 who had been involved in ongoing discussions about the ToC. The stakeholders provided feedback and suggested changes to the ToC. They agreed that the ToC largely represented the inputs from the initial workshops.

## Results

### The CIC ToC

The ToC (Figure 1) is organized into three rows which represent interdependent pathways to impact and four temporal phases. By pathways, we are referring to a temporally linked set of outcomes and related activities which lead to the impact and broadly represent the ecological levels of the system. These three pathways are (1) *community organization and leadership*; which outlines the organizational and systems level changes which need to be in place to provide (2) *community care*; whereas (3) *inclusion and normalizing mental healthcare* outlines the changes required in the broader community to support community care. The four horizontal temporal phases are (1) planning and engaging, (2) building capacity, and systems; (3) implementing the initiative and (4) long-term outcomes. The vertical line on the right side of the ToC indicates the ceiling of accountability which is the point at which the initiative is no longer

directly responsible for the outcomes. The ToC contains 10 strategies (Figure 1) which we hypothesize will lead to the outcomes. Below we describe the underlying principles, outcomes, strategies and the assumptions included in the ToC.

### Guiding principles of CIC

The ToC process and resultant ToC map helped to solidify the concept of CIC as an initiative which is embedded in the community and implemented by local implementing organizations (including community-based organizations [CBOs] and the private sector) who provide *community organization and leadership.* These implementing organizations support CIC helpers who are trained and supported to provide evidence-based *community care* to people in the communities in which they live and work. In addition, CIC helps to support the *inclusion and normalization of mental health.* The guiding principles underlying CIC are that CIC:

1. Reframes mental health as a continuum using a dimensional approach that goes beyond a narrow diagnostic approach of illness.
2. Is an inclusive approach to democratizing and empowering individuals to learn how to respond to mental health problems and addiction issues and take helpful action in the moment.
3. Is based on evidence for task-sharing adapted to diverse contexts in the United States.

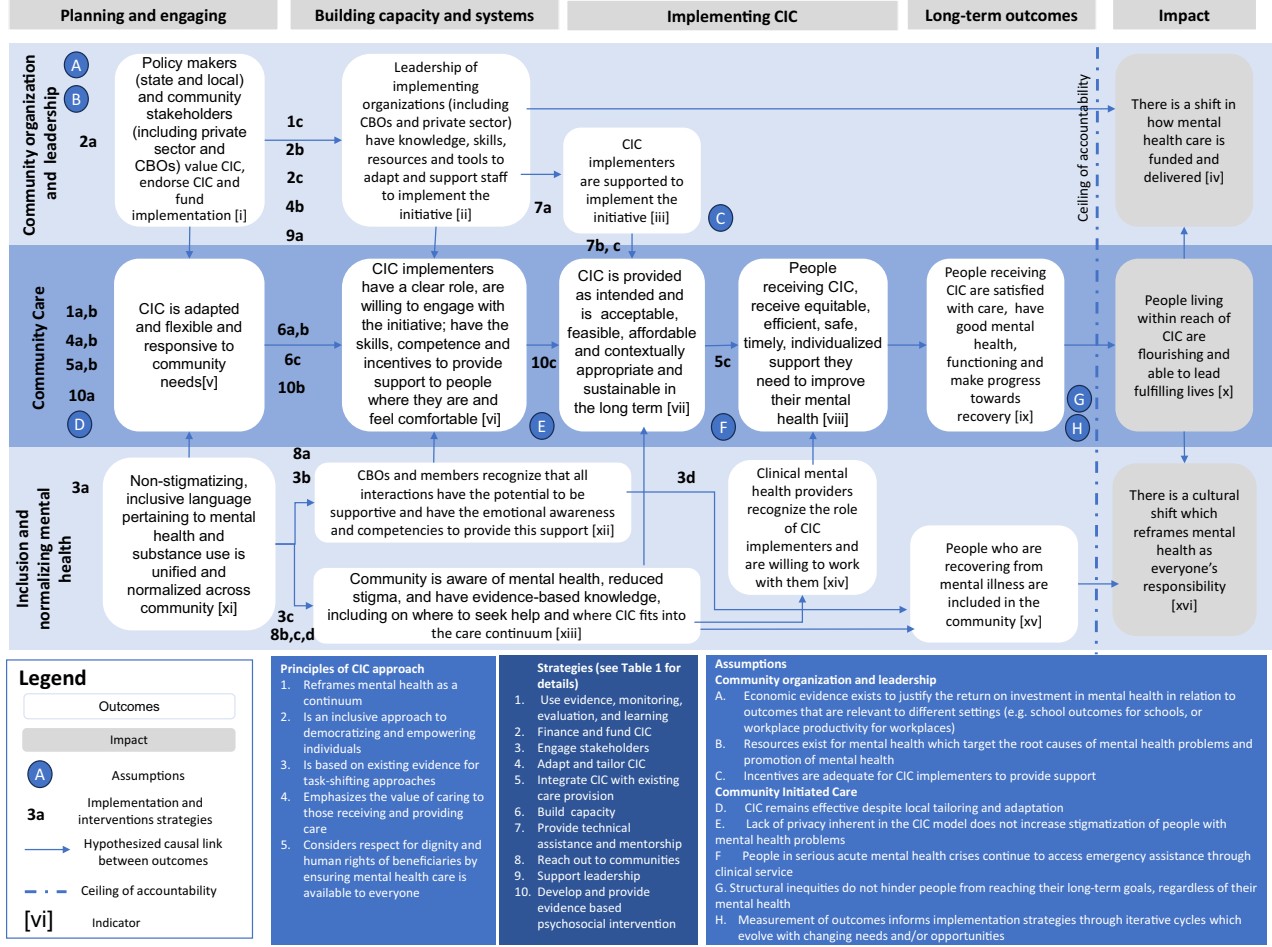

**Figure 1.** Community Initiated Care theory of change.

4. Emphasizes that the act of caring and supporting others has an evolutionary imperative and therefore is beneficial to both the giver and receiver of support (Kohrt et al., 2020)

5. Respects the dignity and human rights of beneficiaries by ensuring mental health care is available to everyone with a specific focus on equity, diversity and inclusion to support all people across age, race, ethnicity, gender, disability and those who experience disparate mental health outcomes (including people with serious mental illness).

## Pathways to impact

The overall impact that CIC is aiming to realize is to ensure that *people living within reach of CIC are flourishing and able to lead fulfilling lives* through three interdependent pathways to impact: (1) community organization and leadership; (2) community care and (3) inclusion and normalizing mental health. An additional two distal societal impacts which CIC aims to contribute to are to promote (1) *a shift in how mental health care is funded and delivered* and (2) *a cultural shift which reframes mental health as everyone's responsibility*. In the ToC, these impacts are beyond the ceiling of accountability. This means that although CIC will aim to influence these impacts, we recognize that the complexity of human interactions, the prevalent social, economic and health-related determinants of mental and other factors which influence these impacts and are beyond the control of CIC. We describe each of the three pathways below.

1. *Community organization and leadership* focuses on the organizational and leadership required CIC to be implemented. Specifically, in the *planning and engaging phase*, CIC requires buy-in from a range of stakeholders in the community, including leadership at state agencies and the local level, large employers, insurance carriers, health systems, community-based organizations and the communities themselves. These stakeholders need to value the program and provide funds to support implementation and scaling up. During the *building capacity and systems phase*, the leadership of implementing organizations such as CBOs (including schools, housing programs, community clinics) or the private sector needs the knowledge, skills, resources and tools to adapt and support staff to implement the initiative. During the *implementation phase*, CIC helpers need to be supported to implement the initiative.

   The key strategies include conducting a landscape and legal analysis to understand the evidence and the legal implications of CIC and working with funders and decision-makers to identify financing for the CIC. We identified the need to engage with stakeholders, including clinicians, about the role of CIC helpers and how they will complement the existing services available. We highlighted the importance of creating a technical hub for support, supervision, technical assistance (including evaluation) and mentorship as well as leadership training.

   Several assumptions related to this pathway were articulated, notably that incentives are adequate for CIC helpers to provide support, that economic evidence exists to justify the return on investment in relation to outcomes that are relevant to different settings (e.g., educational outcomes for schools, or workplace productivity for workplaces) and that resources exist for mental health which target the determinants of mental health.

2. *Community care* focuses on the development and implementation of the CIC helper role in the community. Specifically, in the

*planning phase* in each local context, the CIC needs to be adapted, flexible and responsive to community needs. In the *building capacity and systems phase*, the CIC helpers need to have a clear role, need to be willing to engage in the initiative, have the competence and skills to provide support and have the incentives to provide support to people where they feel comfortable. In the *implementation phase*, there are several interrelated outcomes: that CIC is provided as intended; that it is acceptable, feasible, affordable, sustainable and contextually appropriate for people in need of support; and that for people receiving this care, it is equitable, efficient, safe, timely, individualized support they need to improve their mental health. These lead to the *long-term outcomes* where people receiving CIC are satisfied with the care, have good mental health and functioning and make progress toward recovery.

The strategies in the community care pathway include reviewing existing evidence and developing a for monitoring, evaluation and quality improvement; adapting the CIC in local communities and create-setting specific CIC models; integrating CIC within existing care, including mapping out existing services which can complement CIC and referral to formal clinical care; the development of a tiered set of evidence-based skills and competencies which will be used to equip CIC helpers with the requisite knowledge and skills; and linking the acquisition and maintenance of skills into a learning pathway for CIC helpers. In addition, we need to develop evidence-based psychosocial interventions which can be provided flexibly with respect to time and place in line with the needs of the community.

Several assumptions were identified within this pathway to impact. These include that CIC remains effective despite local tailoring and adaptation; that people in serious acute mental health crises continue to access emergency assistance through clinical services; and that structural inequities may make it difficult for people to reach their long-term goals.

3. *Inclusion and normalizing mental health*: The third pathway focuses on the inclusion and normalizing of mental health in the community so that it is recognized as an integral component of our individual and interpersonal experience. In the *planning phase*, we need to develop nonstigmatizing and inclusive language related to mental health. In the *building capacity and systems* community members need to have the emotional awareness and competencies to provide support in everyday interactions, in particular for people living with mental health problems; and knowledge of the CIC and how it fits into the care continuum. In the *implementation phase*, clinical mental health providers need to recognize the role of CIC providers and be willing to work with them. The *long-term outcome* is the inclusion of people recovering from mental health problems in the community. Two key strategies are included in this pathway. The first focuses on engaging stakeholders through convening stakeholders to inform language used in CIC and hosting dialog in community settings, including about the role of CIC helpers. The second includes strategies related to reaching out to communities, including by developing communication toolkits and other strategies to reach specific populations within the communities, and the implementation of these strategies. No assumptions specific to this pathway were identified.

**Table 2.** Example of CIC indicators

| | ToC outcomes | Construct | Level of data collection and analysis | Type of data | Measurement instrument(s) | Process indicator |
|---|---|---|---|---|---|---|
| Indicator # | ToC outcome | Construct | Data source | Data collection | Measure | Additional process indicators |
| iii. | CIC helpers are supported to implement the initiative | Implementation leadership | Organization provider | Survey or interview | Implementation leadership scale (Aarons et al., 2014) | Percentage of CIC helpers retained after 1 year |
| vi. | CIC helpers are able to provide support to people where they are and feel comfortable | Feasibility | Individual consumer | Survey or interview | FIM (Weiner et al., 2017) | Average number of clients each CIC helper Number of diverse settings where CIC is being actively implemented |

CIC, Community Initiated Care; FIM, feasibility of intervention measure; ToC, theory of change.

## *Indicators*

The ToC also provides a structure for an evaluation framework. Each outcome is linked to one or more measurement constructs for which we developed indicators. For each indicator, we specified the level of data collection and analysis, type of data and data source, potential measurement instrument and a process indicator. Table 2 outlines some examples of how this was operationalized. This will be developed further during the next phase of the initiative.

## Discussion

Despite the significant resources being devoted to mental health care in the United States, the mental health care system is fragmented, overburdened and underfunded and has failed to shift the needle on the burden of mental health problems in the country. A variety of supply- and demand-side barriers, ranging from inequitable access and cost to stigma related to mental health and substance use issues, contribute to the growing and unmet need for mental health care. We propose CIC as a novel inclusive initiative underpinned by a dimensional perspective of mental health and the rich research testifying to the effectiveness of task-sharing to mobilize, equip and empower community members to support the mental health of people in their community.

Informed by implementation science frameworks and models, we have used the participatory ToC development process to understand what is needed to implement the CIC. Intervention and context-specific program theories (including Theories of Change, program logics or logic models) are important to understand how the specific intervention(s) and their implementation strategies are likely to result in an impact. This combines the knowledge from implementation science, the research evidence from other similar interventions collated through evidence synthesis and professional knowledge, the lived experience of people with mental health problems as well as specifying program-specific outcomes and impacts.

This participatory ToC development process presented in this article is a first step in operationalizing CIC and we expect that this ToC will guide the further development and implementation of the CIC. The ToC identifies what is required in three pathways, for *community organizational and leadership*, in the provision of *community care* as well as within the community to ensure *inclusion and normalization of mental health* so that we can contribute to people in reach of CIC are flourishing and able to lead fulfilling lives. The key next step for the CIC is operationalizing the psychosocial interventions and the wide-ranging implementation strategies which include stakeholder engagement, technical assistance and mentorship. This will be done using evidence from other initiatives captured in our landscape analysis, as well as the evidence emerging from global mental health and implementation research (Proctor et al., 2009; Wagenaar et al., 2020; Singla et al., 2021). Initially, the focus will be on developing and implementing low-intensity psychosocial interventions delivered to people with mental health needs, by those who are already engaged in activities within community-based settings (Kohrt et al., 2023). This will follow the steps of our established methodology used in other task-sharing interventions (Patel et al., 2022). We anticipate a suite of hybrid curricula, comprising elements drawn from established evidence-based interventions, along with diverse but evidence-based content tailored to the needs of local communities. In the *community organization and leadership* pathway, we expect developing leadership capabilities to be a critical ingredient for success. This includes increasing understanding of the benefits of CIC among federal and state government officials to facilitate a favorable policy and regulatory environment, equipping community-based organizations with the necessary tools to take a leadership role for CIC implementation and engaging local government leaders in mobilizing resources for the scale-up, catalyzing wider adoption and uptake by diverse stakeholders, and sustaining the initiative in the long-run. Our teams have begun work on some of these deliverables, for example preparing a blueprint for a single session encounter for promoting mental health and supporting persons in distress based on psychological and social science evidence and designing a new executive education program (CHAMPIONS) to build leadership capacity. Then CIC will be adapted to each context building on existing work from the United States and elsewhere (Siddiqui et al., 2022), for example tailoring it to specific delivery platform such as schools, and including additional stakeholders such as the police, justice and prison systems or teachers and school administrators. The implementation and evaluation of CIC will help to provide further evidence for targeted prevention by community members.

We acknowledge certain limitations in our study. Firstly, the research was conducted within the first 2 years of the pandemic, which led us to only include participants who could engage digitally. Secondly, due to the desire for in-depth discussions, we had to limit the number of stakeholders involved in the ToC workshops. Our focus on diversity primarily centered on incorporating a range of perspectives and experiences to ensure equitable care, involving participants from organizations representing diverse stakeholder groups who may face disparities in accessing care. However, we did ensure that we include people who have lived experience of mental

health and/or alcohol and substance use needs. However, we did not document the demographics of individual participants, as our main emphasis was on the diversity of the organizations. Additionally, we were mindful of the potential harm or risk posed to historically marginalized communities when asked to share their mental health struggles for a research project without sufficient evidence that it would directly benefit them. Therefore, we explicitly chose not to seek first-hand perspectives from marginalized groups experiencing mental illness for this initial step to develop a ToC. We also acknowledge that while stakeholders from various community organizations were included, we had limited representation from other sectors such as the criminal justice system, police and school system. We also noted that the stakeholders were predominantly from urban settings and democratic-leaning parts of the country, lacking broad geographic representation across the United States. However, it is important to note that this initial ToC was not designed to focus attention on community dynamics, including geographical factors. Conducting further ToC workshops in each implementation community will allow us to identify and address differences in community needs, resources and capacity, which are best understood and identified by the members of each specific community. Moving forward, as we plan for the implementation of CIC, we are committed to developing context-specific ToC in collaboration with a more diverse and representative group of stakeholders. This ensures that the initiatives we develop are tailored to the specific contexts and requirements of each community.

The future of mental health is local. CIC has the potential to augment our current system's ability to fully respond to population needs by equipping community members with the skills to leverage the largely untapped resource of our daily interactions to recognize and provide support to those in need. Through this locally tailored approach to caring for those around us, we believe that communities can directly contribute to reducing the prevalence of mental health and addiction concerns, preventing their escalation, reducing the overall unmet need for care and addressing associated inequities. Many have argued that the United States mental health care system is in a crisis: now is the time for something radically different for mental health.

**Open peer review.** To view the open peer review materials for this article, please visit http://doi.org/10.1017/gmh.2023.49.

**Data availability statement.** All data generated or analyzed during this study are included in this article.

**Acknowledgments.** We gratefully acknowledge the participants in the theory of change workshops and Well Being Trust for funding support for this work.

**Author contribution.** B.F.M, V.P., S.S. and S.B. conceived the study. A.M., J.A.N., L.B. and E.B. developed the protocol with feedback from V.P., S.B. and S.S. E.B., L.B., A.M., R.R., M.P. and J.A.N. conducted the Theory of Change workshops and developed the Theory of Change. E.B. and A.M. wrote the first draft of the manuscript with critical input from all authors. All authors reviewed the final draft of the manuscript.

**Financial support.** This research was supported by Well Being Trust (which has since been closed) and employees of the Well Being Trust at that time form part of the authorship team (L.B., M.P., R.R., B.F.M.).

**Competing interest.** The authors do not have any conflicts of interest.

**Ethics standard.** Ethical approval was obtained from the Harvard Medical School Institutional Review Board.

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
