## [Reviewer Report]

This article is an important and timely call to action. As the authors articulate, there is a pressing need for community led mental health care in the United States, a strong evidence base for the effectiveness of community led approaches elsewhere in the world, and an increased awareness of the need to address the growing mental health care crisis. The article’s proposed approach to tackle this challenge, the Community Initiated Care initiative, includes a number of unique and important features which increase the likelihood of its success. However, there are certain components that would also benefit from further emphasis within the TOC (minor revisions).

STRENGTHS

- Dimensional and transdiagnostic approach to mental health.One of the strengths of the CIC approach is that it adopts a transdiagnostic and dimensional approach, which recognizes mental health as a continuum. However, this may also pose a challenge when it comes to evaluating the effectiveness of the initiative or related interventions. Measurement tools and scales for mental health are often based on categorical criteria, and thus there may be a need to develop or adapt new tools for measuring impact at the individual level (assuming this is a goal of the CIC).

- An emphasis on leadership and training of leaders. The focus on the importance of developing leadership capacity as part of this process is a unique contribution of the CIC approach. This priority could capitalize on the growing number of strengths based leadership programs targeting educators, policy makers and members of industry around the world (e.g. Global Leadership Initiative, PRIMED, among others).

- Good TOC methodology and wide reaching consultation.The methodology for developing the TOC was thorough and aligned with best practices. The stakeholders involved in the development process included a wide range of individuals with lived experience as well as clinicians, community leaders, policy makers, and funders.

RECOMMENDATIONS

- Better situate the challenge of mental health care in the US within the global context. In the opening pages of the article, the authors share some key statistics about the state of mental health care in the US (e.g. percentage of people requiring and accessing quality care). It would be helpful to provide the reader with a point of reference for these statistics, by comparing the situation in the US with the global one, or comparing it to countries of similar population, economy, etc. Likewise, an indication of the amount of spending on mental health care in dollars would help to illustrate the need for a more cost effective approach. Finally, there is an argument that community care is more cost-effective - are there examples from other places that have adopted such a model to be used to back up this claim?

- Acknowledge and involve actors from the judiciary and prison systems as key stakeholders in the development and implementation of the CIC. The CIC is by definition a multi-sectoral approach that will require the support of diverse stakeholders. The article/TOC acknowledges the importance of policy leaders, community based organizations, and clinicians/formal health sector professionals and also makes reference to faith and education leaders. However, for the CIC to be a success it will be essential for the initiative to engage the support of the police, justice and prison systems, particularly given the high levels of incarceration for individuals with mental health challenges and the importance of their rehabilitation into the community. This group of actors should be explicitly acknowledged in the initiative description, and iIt would be beneficial for any further revisions to the TOC to include stakeholders from this group also.

- Ensure a solid understanding of the extent of formal and informal community mental health care provision and training initiatives underway prior to implementation. While the US mental health care model is clearly dominated by professional and clinical practice, there is a certain amount of informal, community based care that is already being provided via faith and cultural leaders. Likewise, there are school based initiatives to provide SEL or leadership training, especially within the charter school system in the US that can be built upon. It will be important for the initiative to map and understand this landscape and engage the relevant actors before implementing the CIC, and it would be good to see this step more explicitly outlined in the TOC.

- Address the potential for burnout among community health providers by providing requisite support and training. The CIC approach recognizes the evolutionary imperative to provide care, and the potential benefits of caregiving for individuals providing care and community cohesion. This is an important and often overlooked benefit. However, it will also be important to acknowledge and address the potential for burnout among community providers, and to ensure that they are adequately supported. Burnout is an important concern in many countries that rely on CHWs to deliver mental health care, such as the ASHAs in India and the LHWs in Pakistan. The CIC approach should acknowledge and address this challenge.

- Elaborate on the process for identifying and selecting locally appropriate interventions that individuals will be trained on. Communities in the United States are heterogenous and highly diverse. The success of the CIC will rely on its ability to develop a coherent approach to engaging and training community providers to deliver interventions that are locally relevant, context appropriate, and evidence based. Furthermore, it will be important to develop resources and tools to enable communities to select, develop or adapt such interventions based on specific needs. This selection and adaptation process is critical and should be an explicit step in the TOC. As part of the suite of resources offered by the CIC, the authors may want to consider developing a database, menu or playbook of interventions that communities can consider (the MHIN site, which highlights diverse interventions could serve as a starting point for this resource). Given the important evidence base for community care in LMICs, and the fact that the US is home to significant numbers of immigrants from around the world, this resource could also incorporate community based interventions that have been developed and implemented outside the United States.

---

## [Reviewer Report]

Thank you for the opportunity to review this manuscript; incorporating opportunities for community-based mental health care in low-resource settings in the US is important and this manuscript describes a process for getting local stakeholder perspectives and buy in to initiate this process. To support the interpretation and impact this work can have for a wider-audience, I suggest the following major revisions:

Overall Comments:

- In its current form, the manuscript reads more like a report rather than research manuscript; please include defined Methods, Results, Discussion sections as well as limitations of the approach.

- The authors use a range of terms for mental health and appear to use them interchangeably at times: mental illness, mental disorder, mental health problems, mental health needs, psychosocial needs, and recovery. It would be helpful for the authors to be consistent and then provide definitions when different terms are used. It is also sometimes confusing to understand whether the authors are promoting the CIC as a model for mental health treatment, prevention and/or promotion.

- The authors rightly acknowledge that access to care is worse for many historically disadvantaged groups. It would be helpful to understand if members of these groups were part of the initial 24-person stakeholder or follow up 10-person stakeholder groups and how their perspectives were incorporated in the TOC.

- The description of the TOC process suggests agreement among the stakeholders; it would enrich the manuscript to note where particularly difficult discussions came up and where areas of disagreement were identified.

Specific Comments:

- A table providing basic demographic characteristics (gender, race/ethnicity, role) for the stakeholders would be helpful to better understand who was at the table.

- The authors define “task-sharing” as basically synonymous with “task-shifting” – however, the former specifically includes a relationship among the providers with others to “share” the task of caring for the service user. It is unclear if the TOC model is suggesting use of “task-shifting” or “task-sharing”.

- The authors cite recent systematic reviews of task-sharing for mental health problems in LMIC settings. This is accurate, but these intervention trials were for treatment-based interventions (i.e. to treat people with high depression scores or diagnosed depression) rather than single-session psychosocial programming to prevent distress or disorder or promote well-being. The authors seem to be promoting the idea of single-session prevention/promotion programming, so providing evidence for the acceptability and effectiveness for those type of intervention models is needed.

- In the section entitled “A Theory of Change for Community Initiated Care”, the authors note the importance of identifying the “assumptions or conditions which need to be in place”. This seems quite important for other researchers to better understand if they want to replicate this approach. It would be helpful for the authors to provide examples of the assumptions and conditions that came up in the TOC discussions to allow for readers to understand which may be similar to the contexts in which they are working.

---

## [Reviewer Report]

Initial comments

The aim of the paper is to “describe the methods and results of a Theory of Change workshop with stakeholders which illustrates the hypothesized pathway for how we might advance CIC [Community Initiated Care] in the United States…”

The paper’s methods include a series of two workshops with a purposive sample of stakeholders. Researchers iteratively developed a Theory of Change based on input from the first workshop. Then they presented that draft to stakeholders for feedback during the second workshop.

Appreciation to the authors for drawing important attention to pressing mental health needs in the U.S., and the promise of deploying task-sharing methods to meet those needs. It highlights the need to import ideas from around the world, particularly from low-and-middle-income countries, on how to use community, non-specialist approaches to provide mental health support.

Introduction

1. The introduction articulates the myriad of challenges in the US mental healthcare system.

2. It introduces important key concepts used in the paper.

3. The title includes the term “scale”. The focus of the paper seems to be more about Theory of Change. I wonder if “A theory of change for community initiated mental health care in the United States” might be a tighter fit with the content of the paper.

4. Pg. 3. Line 71 – I wonder if the introduction could be strengthened by defining and more clearly differentiating the terms mental health and mental illness. This would also be in service of the later discussion about the continuum of mental health in the introduction.

5. Pg. 3 line 73 – are the authors conceptualizing “mental health care” as including medication? Or are they focusing on CICs that would include non-medication therapies (e.g., talk therapies)? I ask because throughout the paper it seems like candidate CIC interventions would be talk/behavioral/experiential therapies. Within the broader umbrella of mental health care, medication is the most frequently used service in the U.S. And task-shifting prescription rights in the U.S. would involve larger changes that, currently, may not be fully accounted for in the ToC. Though, this has happened to an extent with Physician Assistants and Nurse Practitioners.

6. Pg. 3 line 70 to pg. 4 line 90 – I think the broader point that many don’t receive care, even more so for individuals racialized at BIPOC, is valid and important to explain. I wonder if the current draft of this paragraph may be oversimplifying the mental healthcare system. For instance, most mental health services in the U.S. are provided by primary care doctors in the form of psychotropic medication prescriptions. Outside of medication, most talk therapies are provided by master-level clinicians. All but one state has peer specialist certifications, with most states offering reimbursement for those services. There are programs in the US who use community health workers to deliver low intensity interventions, schools are deploying universal prevention curricula. Certainly, there is a lot of room for growth and improvement still. Where do the authors envision CIC fitting in the existing landscape? Is the issue that there needs to be more task-sharing? Different task-sharing? Organized differently?

7. Pg. 4 line 79 – great point about the density of providers compared to other parts of the world. It would be great to add a citation here.

8. Pg. 4 line 92 to pg. 5 line 105 – I think it is great that the authors are speaking about the dimensionality of mental health, such an important concept! Tying back to an earlier comment about differentiating the terms mental health and mental illness, is this section referring to mental illness? Dr. Corey Keyes has written several papers on the topic of mental health dimensionality and has developed two continua: one for mental health and the other for mental illness. The mental illness continuum has no mental illness (i.e., normative reactions) on one end and serious mental illness on the other. This seems most closely aligned with the dimensional approach referenced in the paper. Keyes also introduces the concept of a continuum of mental health (separate from mental illness). That continuum spans from languishing (poor mental health) to flourishing (optimal mental health). I wonder if this section could be a little clearer by either specifying mental health or mental illness (currently seems mental illness focused), or expanding the paragraph by bringing in additional content about both continua?

9. Pg. 6 line 128 – Task-sharing is such an important approach, appreciation for mentioning it. Do the authors view CIC interventions to be primarily task-sharing approaches? Are there other examples of CIC that don’t involve task-sharing?

10. Pg. 6 134 to 146 – I enjoyed reading about the benefits of community care. The previous paragraph seems to say, “task-sharing works” and this paragraph seems to focus on “this is why we think it works”. I wonder if the first sentence of this paragraph might be changed to highlight that focus? I think the terms “implementation science” and “scaling up” might detract from the paragraph’s purpose.

11. There is literature in implementation science and scale-up about: determinant frameworks, process models, implementation guides, indicator/outcome operationalization and measurement, and logic models. The authors are engaged in an important project. I wonder if the authors could strengthen their case in the introduction about why an additional model is needed. There are critiques in the literature that suggest more models may not be helpful. I wonder if the authors might address that concern so readers will know how to best use the paper.

Methods

1. I’m not familiar with this journal’s formatting requirements. The readability of this paper would be enhanced for me if there were headings separating the introduction, methods, results, and discussion sections.

2. Pg. 8 line 177 – how were the participants identified?

3. Pg. 8 – the generation of the inputs into a ToC seems to rely on qualitative methods. Did the authors use one of the qualitative reporting guidelines to draft this paper (e.g., COREQ)? It could help guide the authors on information to include that would increase the utility of the paper.

4. Pg. 8 – what artifacts/inputs were gathered from the participants to be analyzed by the research team? If verbal feedback was used, how was it captured and accuracy ensured (e.g., sessions were recorded and transcribed)?

5. Pg. 8 – what method did the researchers use for the thematic analysis? How were the themes generated? Who was involved in coding and/or synthesizing? Various qualitative techniques can be used to ensure the trustworthiness of the findings. What techniques did the authors use?

6. Pg. 8 line 196 – that’s great that the authors are considering the contributions that implementation science can make on this project. What facet of implementation science did the authors review to inform their work? The cited Proctor article is a wonderful resource and presents one framework. Is that the only framework that was reviewed? A recent count identified 114 possible implementation science frameworks. Most studies within implementation science will use one or two frameworks. More detail would be helpful about why this framework was the best fit for this study. There is guidance on how to select a framework for use in studies that could help (e.g., Birken et al., 2017; Birken et al., 2018; Moullin et al., 2020). The authors might replace the term “implementation science” to the name of the framework they used.

7. Pg. 9 line 199-208 – the terms “implementation strategies”, “interventions”, “components”, and “activities” seem to be used interchangeably in this section. I wonder if this section would be clearer if these terms were defined.

8. Pg. 9 line 199-200 – There is a sizeable literature on implementation strategies (identification, selection, development, reporting, etc.). What strategy taxonomy did the authors use? How was the initial pool of strategies created? What procedures were used to translate workshop participant feedback into strategies? Were all participant reported strategies included? If not, what inclusion rules were used? The authors signal the use of implementation science in a few places in the paper. Implementation science is great and has a lot to offer this project. It is unclear in the reporting of this paper what areas of implementation science were used (e.g., frameworks, strategies, measurement, mechanisms, etc.), the extent to which those areas, and which procedures they relied on.

Results

1. Pg. 9 line 220 – I’m having a hard time understanding what is meant by the term “pathway”. Are these representing ecological levels of the system? Actors within the larger system? Pathway 1 and 2 seem to be labeled as actors, but the 3rd seems more like a broader cultural piece. Would the actor in pathway 3 be community members outside the service delivery system? Pathway 1 didn’t have a long-term outcome. If there isn’t one, does it make sense to conceptualize this ToC with separate pathways? Is there mostly one pathway, community care, with actors that support those efforts?

2. Pg. 11 line 249 –the overall impact sentence doesn’t match the figure. The figure has that impact attached to the second pathway.

3. Pg. 11 line 270 – referencing table 1 would help here.

4. Pg. 12 line 274-281 – I’m having a hard time knowing how the assumptions fit with the other components of the ToC. Which phase of the ToC do the assumptions apply to, and what is their relationship with the strategies/interventions? For example, is assumption A under community organizations saying that strategy 2 worked? If so, would it be more parsimonious to modify the strategy to say, “adequately finance and fund CIC”? When is it better to move an assumption to a target for action, outcome, or impact? For example, assumption C under community organization could be a long-term outcome. Or it could be an action (i.e., find/generate resources to target root causes…)

5. Figure (based on the draft I have):

o Several arrows are not linking to anything and at times appear in the middle of the text.

o Some text doesn’t fit within the squares.

o Long-term outcome is missing for pathway 1.

o Pathway 1 and 2 have text boxes between the building capacity and systems column and the implementation the initiative column. Should they be applied to one or the other?

o For me, the implementation strategy numbers were initially confusing. It became much clearer when I read table 1. I wonder if it would be better to remove the implementation strategies box from the figure. Instead, number all the strategies in the table (1a, 1b, 1c, 2a, 2b, 2c, etc.) and then use those numbers to the figure? If the intention is to use all strategies under each domain (e.g., if 1 means 1a, 1b, and 1c will all be used), then the authors wouldn’t need to specify 1a, 1b, 1c. The difficulty I had was that the description of the strategy clusters didn’t immediately make sense to me in relation to where they were placed. Then when I saw the more specific strategies in the table, I understood.

o Table 2

□ Add a column to match the numeral from the figure.

□ Is acceptability the best fitting construct for the second row? Would willingness be closer?

Discussion

1. Pg. 14 line 338 – perhaps replace “implementation science” with research.

2. Pg. 16 line 377 – I love the line about the future of mental health being local!

3. Similar to my comment in the introduction, the discussion section could be strengthened by showing how this ToC adds to the literature. The authors are engaged in an amazing project and it’s clear that the ToC is helping advance their work. What is less clear to me is why readers should use this ToC over existing implementation science or scale-up frameworks. Both of those disciplines have articulated determinants and processes to implement or scale interventions. CIC could be inserted as an intervention in those frameworks. What does this ToC add/improve/clarify/change?

---

## [Reviewer Report]

The authors have appropriately addressed the key comments from the external reviewers. The manuscript has been strengthened by improvements to the organization of the paper, definitions of key terms, expanded information about the CIC model itself, references to additional stakeholder groups, and additional details about the TOC process as well as who was included in the process and related limitations.

There are some sections in the tables (see p. 28 onwards) where text is truncated or cut off, but this is a formatting issue rather than a substantive one.

---

## [Reviewer Report]

Thank you for the revisions, many of my initial concerns and queries have been addressed. A couple remaining points would benefit from further clarity:

- It would be help to clarify up front if there may be differences in the role of this approach for severe mental illnesses, particularly those with more of a biological basis such as schizophrenia and bipolar disorder. In the introduction the authors’ state

“This dimensional approach views many mental illnesses as extreme and stressful versions of common, normative, human experiences (for example, sadness, loneliness, grief, despair)(Whitley and Drake 2010) which create sustained dysfunction over time if not addressed”. While this is accurate for many common mental health problems that fit in the depressive/anxiety domains – it may be less accurate for conditions with psychotic symptoms for example. Further clarification as to where this community approach may be best situated in the care system for the full spectrum of mental health conditions would be helpful.

- The evidence for task-sharing interventions described in the background and referenced in the discussion is strong for treatment interventions but is rather weak for psychosocial programs that focus on prevention and promotion, which seem to be the planned focus of the CIC next step activities. Acknowledgement of this gap would be helpful – and as well as addressing whether the CIC approach could serve as an opportunity build the evidence base for community psychosocial prevention programming.